# Long-Term Observational Outcomes after Total Correction of Congenital Heart Disease in Korean Patients with Down Syndrome: A National Cohort Study

**DOI:** 10.3390/children9091329

**Published:** 2022-08-31

**Authors:** Ji Hee Kwak, Seung Won Lee, Hye Ryeong Cha, June Huh, I-Seok Kang, Tae-Gook Jun, Ji-Hyuk Yang, Man Yong Han, Jinyoung Song

**Affiliations:** 1Department of Pediatrics, Kangbuk Samsung Hospital, Sungkyunkwan University School of Medicine, Seoul 03181, Korea; 2Department of Data Science, Sejong University College of Software Convergence, Seoul 05006, Korea; 3Department of Precision Medicine, Sungkyunkwan University School of Medicine, Suwon 16419, Korea; 4Department of Computer Science and Engineering, Sungkyunkwan University, Suwon 16419, Korea; 5Department of Pediatrics, Adult Congenital Heart Clinic, Heart Vascular Stroke Institute, Samsung Medical Center, Sungkyunkwan University School of Medicine, Seoul 06351, Korea; 6Department of Thoracic and Cardiovascular Surgery, Samsung Medical Center, Sungkyunkwan University School of Medicine, Seoul 06351, Korea; 7Department of Pediatrics, CHA Bundang Medical Center, CHA University School of Medicine, 59, Yatap-ro, Bundang-gu, Seongnam-si 13496, Korea; 8Department of Pediatrics, Samsung Medical Center, Heart Vascular Stroke Institute, Grown-Up Congenital Heart Clinic, Sungkyunkwan University School of Medicine, 81 Irwon-ro, Gangnam-gu, Seoul 06351, Korea

**Keywords:** down syndrome, congenital heart disease, surgical outcome, total correction

## Abstract

Background: In the present study, the population prevalence and postoperative morbidity and mortality in Down syndrome patients who underwent total correction for congenital heart disease were investigated using data from a large national cohort. Methods: Retrospective administrative data from 2,395,966 participants born between 2008 and 2012 were acquired from the National Investigation of Birth Cohort in Korea. Among Down syndrome patients, 58.3% had congenital heart disease and 32.3% underwent total correction. Propensity score matching (maximum 1:1) and stabilized inverse probability treatment weighting (IPTW) were performed for each group (153 Down syndrome patients and 4482 non-Down syndrome patients). Results: T late mortality rate was significantly higher in the Down syndrome group than in the non-Down syndrome group (8.1% vs. 3.8%). No differences were observed in postoperative heart failure and arrhythmias, but pulmonary hypertension was significantly greater in the Down syndrome group than in the non-Down syndrome group (26.9% vs. 7.0%). The length of hospitalization was longer in the Down syndrome group than in the non-Down syndrome group (14 days vs. 11 days; interquartile range (IQR): 10–25 vs. 6–19; *p* < 0.0001). After total correction, readmission frequency for any reason was minimally but statistically significantly higher in the Down syndrome group compared to the non-Down syndrome group (5 times vs. 5 times; IQR: 3–8 vs. 4–9; *p* < 0.0001). However, the number of emergency room visits was minimally but significantly lower in the Down syndrome group compared to the non-Down syndrome group (2 visits vs. 2 visits (IQR): 2–7 vs. 1–4; *p* = 0.016). Conclusions: Down syndrome patients with congenital heart disease undergoing total correction showed pulmonary hypertension after surgery, longer length of hospitalization, frequent hospitalization after surgery, and a higher rate of late mortality.

## 1. Introduction

Congenital heart disease is frequently diagnosed in patients with Down syndrome. Approximately 50% of infants born with Down syndrome are diagnosed with congenital heart disease, which has a significant effect on morbidity and mortality [1,2]. Among the most common cardiac anomalies, atrioventricular septal defect (AVSD) accounts for approximately 45% and ventricular septal defect (VSD) for 30% [1], followed by atrial septal defect (ASD, 20%) and tetralogy of Fallot (TOF, 5–10%) [3]. Improvements in surgical techniques and perioperative management [4] as well as successful early treatment of congenital heart disease are contributing to a significant increase in life expectancy for children with Down syndrome (12 years in 1940 compared to >60 years currently) [1]. In particular, timely surgical correction is important to prevent the development of pulmonary hypertension and Eisenmenger syndrome. However, surgical treatment of congenital heart disease in these patients usually involves a greater risk of postoperative complications and mortality than it does in non-Down syndrome patients [4].

In the present study, population prevalence as well as postoperative morbidity and mortality rates in patients with Down syndrome who underwent total correction for congenital heart disease were investigated using data from a large national cohort.

## 2. Materials and Methods

### 2.1. Data Source and Collection

Retrospective administrative data from 2,395,966 participants born between 2008 and 2012 in Korea were obtained from the National Investigation of Birth Cohort in Korea study [5]. The Birth Cohort study contains information including the National Health Screening Program for Infants and Children and medical utilization obtained through the National Health Insurance Service in Korea. The de-identified private key can be used for research purposes to secure ethical customs clearance in accordance with the current National Health Insurance Act [6]. The Institutional Review Board of the Korea National Institute for Bioethics Policy reviewed and approved the study protocol (P01-201603-21-005).

### 2.2. Study Population

Among 2,395,966 children born in 2008–2012, 4643 with chromosomal abnormalities (Q 91X–Q 99X) other than Down syndrome were excluded. The inclusion criterion was congenital heart disease or congenital malformation of the circulatory system (Q 20X–Q 26X); 70,575 patients were eligible. Among those patients, 5282 who underwent total correction for congenital heart disease were included in the subject group; 234 had Down syndrome and the remaining 5048 were children without Down syndrome. Propensity score matching (maximum 1:1) and stabilized inverse probability treatment weighting (IPTW) were performed and assigned 153 subjects to the Down syndrome group and 4482 subjects to the non-Down syndrome group. The flowchart of participant selection for the study is shown in Figure 1.

### 2.3. Definition of Down Syndrome and Congenital Heart Disease

The National Health Insurance Service data were used to select patients who underwent cardiac surgery for congenital heart disease among children diagnosed with Down syndrome (International Classification of Diseases, 10th Revision (ICD-10) codes: Q 90.0, Q 90.1, Q 90.2, and Q 90.9). Information on congenital heart disease diagnoses and hospital course was coded by trained personnel using the ICD-10 system as shown in Appendix A. For children with more than one congenital heart disease diagnosis, the priority of congenital heart disease was classified as the main diagnosis that affects the complexity and prognosis of the disease.

Surgery type was divided into two groups, palliation and total correction, and the study subjects were patients who benefitted from total correction—either as a single step or after a palliation procedure. Surgical procedure codes for congenital heart disease in patients with and without Down syndrome are provided in Appendix A. The study group consisted of children with Down syndrome who eventually underwent total correction for congenital heart disease.

### 2.4. Outcome Measures

The primary outcomes were postoperative mortality, newly developed cardiac complications, and postoperative clinical course among Down syndrome children and controls with congenital heart disease who underwent cardiac surgery for total correction. Mortality was assessed at 30 days postoperatively; deaths within 30 days after surgery were defined as in-hospital mortality, and deaths after postoperative 30 days were defined as late mortality. The death rate was calculated by dividing the incidence of death by the total follow-up period among Down syndrome and age- and sex-matched controls from 2008–2018. Postoperative cardiac complications such as heart failure, pulmonary hypertension, and arrhythmias were investigated. For consideration of cardiac complications, patients already diagnosed before surgery were excluded, leaving only those newly diagnosed after surgery. Heart failure was recorded using the ICD-10 code I50. Pulmonary hypertension was recorded using ICD-10 codes I27.0 and I27.2. Arrhythmias were recorded using ICD-10 codes I20X, I21X, I22X, and I44X–I49X. The clinical course was investigated based on length of hospitalization, number of emergency room visits, and readmission frequency. Length of hospitalization was defined as the duration of hospitalization in days after total correction. The number of emergency room visits was the sum of all visits to the emergency room for any disease even if not a cardiac complication. Readmission frequency referred to the number of admissions for any diagnosis, even if not a cardiac complication.

### 2.5. Statistical Analysis

Statistical analysis was performed using the SAS System for Windows (SAS Institute Inc., Cary, NC, USA). The IPTW of the propensity score was used to balance the baseline characteristics of the two groups. The propensity score was estimated using multivariable logistic regression with the following covariates chosen a priori: sex, prematurity (yes vs. no), birth weight (continuous value), region of birth (Seoul, metropolitan, city, or rural), income quintile (1 to 5, classified with insurance fee), year of birth (from 2008–2012), type of surgery (total correction or total correction after palliation), and age at surgery; some conditions originating in the perinatal period other than chromosomal abnormalities (Appendix A); type of congenital heart disease (defined in Table 1, Appendix A) [7,8,9]. The patients in the reference group were matched [8]. Differences between groups in baseline characteristics were compared using standardized differences in both the prematched and postmatched samples [10] (differences >10% were considered meaningful). Patients with missing data were excluded from analyses. Weighted risk ratios and 95% confidence intervals (CIs) were obtained using modified Poisson regression [11] for postoperative outcomes, and hazard ratios (HRs) and 95% CIs were obtained using the Cox regression model for mortality. Two-tailed *p*-values < 0.05 were considered statistically significant. Mortality trends were compared using logistic regression analysis, with death as the outcome and independent variables of group, year, and an interaction term between group and year. The Kaplan–Meier method was used to compare survival among patients with congenital heart disease who underwent total correction based on age group and sex using log-rank tests.

Sensitivity analysis was performed using two approaches, while matching the propensity score based on only demographic characteristics or matching demographic characteristics and conditions other than chromosomal abnormality originating in the perinatal period (Appendix A).

## 3. Results

### 3.1. Demographic Characteristics of the Participants

Chromosomal abnormalities occurred in 1243 of the 2,395,966 participants, a prevalence of 5.18 per 10,000 total births. Among them, Down syndrome was the most common chromosomal anomaly, accounting for 21.1% of all chromosomal abnormalities (Q 91X–Q 99X) in patients from 2008–2012. Among Down syndrome patients, 58.3% had congenital heart disease and 32.3% underwent corrective surgery.

Patient demographic characteristics are presented in Table 1. Matching with conditions other than chromosomal abnormalities originating in the perinatal period is presented in Appendix A. Only 17 Down syndrome patients (7.3%) underwent total correction after palliation surgery, and the other 217 Down syndrome patients (92.7%) underwent total correction directly in a single step. The median follow-up period for the participants was 100 months (IQR: 85–115). Missing data were rare (<5% for all variables).

### 3.2. Preoperative Distribution of Congenital Heart Disease

Data on the main diagnosis of congenital heart disease are presented in Table 2. The most common congenital heart disease in the Down syndrome group was VSD, followed by AVSD and TOF.

### 3.3. Type of Cardiac Surgery and Age (Table 3)

In the Down syndrome group, 17 patients underwent palliative surgery before total correction, and 217 patients underwent total correction in one step. AVSD patients not amenable to corrective surgery and being treated with a single ventricle palliation were not included in this series. Having Down syndrome did not predispose patients to undergo more palliation surgeries compared to controls (Table 3). The age at surgery was also not significantly different between groups (Table 3).

**Table 3 children-09-01329-t003:** The type of and age at surgery in patients with and without Down syndrome who underwent total correction.

Surgery	Controls (N = 5048)	Down Syndrome (N = 234)	*p*-Value
**N Based on Surgery Type, n**			
Total correction	4521	217	0.118
Total correction after palliation	527	17	0.118
**Age at Surgery, months** **(IQR) ^a^**			
Total correction	4 (1–9)	4 (2–7)	0.575
Total correction after palliation	9 (3–26)	7 (6–12)	0.458
Palliation	1 (0–3)	1 (0–2)	0.501

N, number of patients; ^a^ Categorical variables were expressed as absolute number and frequency distribution. Continuous variables as median and interquartile range P25–P75.

### 3.4. Postoperative Mortality and Surgical Outcomes (Table 4)

In-hospital mortality between groups was not significantly different. However, late mortality was significantly different, with a higher rate observed in the Down syndrome group than in the non-Down syndrome group (8.1% vs. 3.6%; *p* < 0.0001). The median age of death after total correction was 7.5 months (IQR: 5–25 months) in the Down syndrome group and 5 months (IQR: 2–13 months) in the control group. The cumulative survival rate in Down syndrome patients was significantly lower than in the age- and sex-matched control subjects (HR = 4.3, 95% CI: 1.9–9.9; Figure 2). In adjusted risk analysis, children with Down syndrome had a higher adjusted risk of late hospital mortality after total correction. The common congenital heart diseases in the 26 deceased Down syndrome patients were AVSD (14 patients), VSD (10 patients), TGA (1 patient), and double outlet right ventricle (1 patient). The common congenital heart diseases in the 275 deceased controls were TOF (43 patients), AVSD (42 patients), TGA (42 patients), and VSD (41 patients). After total correction, cardiac complications were investigated. A statistically significant difference was not found in heart failure and arrhythmias; however, pulmonary hypertension was significantly more frequent in the Down syndrome group than in the non-Down syndrome group (26.9% vs. 7.0%; *p* < 0.001). Length of hospitalization at the time of total correction was significantly longer in the Down syndrome group compared to the non-Down syndrome group (14 days vs. 11 days; IQR: 10–25 vs. 6–19; *p* < 0.0001).

**Table 4 children-09-01329-t004:** The postoperative mortality and outcomes for congenital heart disease in patients with and without Down syndrome.

	Observed Data (N = 5282)			Propensity-Matched Data (N = 4635)			Modified Poisson		
	Control	Down Syndrome	*p*-Value	Control	Down Syndrome	*p*-Value	Risk Ratio	(95% CI)	*p*-Value
**Mortality**									
In-hospital mortality (<30 days after operation) *	91 (1.8)	7 (2.9)	0.18	4 (0.0)	1 (0.6)	0.454	3.1	(0.1–70.6)	0.477
Late mortality (≥30 days after operation) *	184 (3.6)	19 (8.1)	0.000	40 (0.8)	6 (3.9)	0.000	4.4	(1.8–10.3)	**0.000**
**Cardiac complication**									
Heart failure ^a^^,^*	901 (17.8)	48 (20.5)	0.299	795 (17.7)	30 (19.6)	0.641	1.0	(0.7–1.5)	0.672
Pulmonary hypertension ^b,^*	355 (7.0)	63 (26.9)	<0.001	296 (6.6)	37 (24.1)	<0.000	3.6	(2.5–5.1)	**<0.0001**
Arrhythmias ^c,^*	672 (13.3)	25 (10.6)	0.245	549 (12.2)	16 (10.4)	0.425	0.8	(0.4–1.3)	0.455
**Clinical course**									
Length of hospitalization ^d^ (at total correction) ^g^	11 (6–19)	14 (10–25)	<0.001	11 (8–18)	15 (10–23)	0.000	1.2	(1.2–1.2)	**<0.0001**
Number of emergency room visits ^e^^,g^	2 (1–4)	2 (1–4)	0.875	2 (1–5)	2 (1–4)	0.072	0.8	(0.7–0.9)	**0.016**
Readmission frequency ^f^^,g^	5 (4–9)	5 (3–8)	<0.001	6 (4–9)	5 (3–8)	0.008	1.6	(1.5–1.7)	**<0.0001**

^a^ Heart failure is recorded using ICD-10 code I50. ^b^ Pulmonary hypertension is recorded using ICD-10 codes I27.0 and I27.2. ^c^ Arrhythmias are recorded using ICD-10 codes I20X, I21X, I22X, and I44X–I49X. ^d^ Length of hospitalization was defined as the days of hospitalization after total corrective surgery. ^e^ Number of emergency room visits refers to the number of visits to the emergency room for any diagnosis, even if not a cardiac complication. ^f^ Readmission frequency refers to the number of readmissions for any diagnosis even if not a cardiac complication. ^g^ Categorical variables were expressed as frequency distribution, and continuous variables as median and interquartile range P25–P75. * Values are numbers (%) unless otherwise indicated. CI, confidence interval; N, number of patients.

After total correction, readmission frequency for any reason was minimally but, in statistical terms, significantly higher in the Down syndrome group than in the non-Down syndrome group (five admissions vs. five admissions; IQR: 3–8 vs. 4–9; *p* < 0.0001). However, the number of emergency room visits after total correction was again minimally but, in statistical terms, significantly less in the Down syndrome group compared to the non-Down syndrome group (two emergency room visits vs. two emergency room visits; IQR: 2–7 vs. 1–4; *p* = 0.016).

### 3.5. Sensitivity Analyses

Sensitivity analysis was performed using two approaches. Even when PSA matching was performed only with demographic characteristics, or with both demographic characteristics and conditions other than chromosomal abnormalities originating in the perinatal period, the approaches showed the same results. Late mortality, pulmonary hypertension, length of hospitalization, number of emergency room visits, and readmission frequency differed between the Down syndrome and control groups.

## 4. Discussion

This study was able to evaluate the outcomes of 234 children with Down syndrome undergoing cardiac surgery. In a previous study [12] in Korea, the prevalence of Down syndrome was 14 per 10,000 total births. In the present study, a Down syndrome prevalence of 5.18 per 100,000 persons in 2008–2012 was observed, a similar prevalence to the 4.4 per 100,000 persons in 2005 and 2006. In a previous study [12], the overall prevalence of Down syndrome in Korea may not have been fully represented due to a limitation of a small number of institutions rather than a nationwide cohort. In addition, prenatal diagnostic testing in Korea may have resulted in an increased abortion rate after a Down syndrome diagnosis, which may have contributed to the smaller number of children with Down syndrome observed in this study.

The congenital heart disease most frequently diagnosed in Down syndrome is AVSD (40%), followed by VSD (30%), secundum ASD (10%), TOF (5%), and isolated PDA (5%) in European countries and the United States [13,14]. Notably, the VSD type of congenital heart disease is more frequent in Asians and Native Americans than in white populations [15]. Recently, surgical treatment of congenital heart defects in Down syndrome patients has been routinely performed [16]. Furthermore, in general, children with Down syndrome reportedly are younger at the time of surgery [17]. Early surgical treatment in Down syndrome can reduce the upper respiratory and feeding/growth problems associated with Down syndrome in addition to the growth retardation and respiratory symptoms associated with congestive heart failure. In particular, concerns for the development of early pulmonary vascular disease in Down syndrome with cardiac defects involving significant left-to-right shunts have played a role in earlier surgical intervention [18]. Approaches to the treatment of Down syndrome patients with congenital heart disease have changed over time. In the past, surgical treatment of heart defects was often not considered due to long-term natural history and decreased life expectancy, and higher perioperative morbidity and mortality were reported [19]. However, life expectancy and treatment for noncardiovascular diseases in patients with Down syndrome, such as respiratory and neurodegenerative complications, have meanwhile improved [20]. Outcomes in patients undergoing congenital heart surgery have also improved due to improved surgical techniques, perioperative management, and younger age at operation [21]. Regarding pulmonary hypertension in the present study, although no difference was observed in time of surgery between the Down syndrome and control groups, the incidence of newly developed pulmonary hypertension after surgery was significantly higher in the Down syndrome group. This is not surprising, since children with Down syndrome and congenital heart disease develop pulmonary hypertension more often than children with congenital heart disease without Down syndrome do [22]. Regarding arrhythmia development after surgery for congenital heart disease in patients with Down syndrome, current evidence shows that the incidence of atrioventricular block after surgery that requires pacemaker placement is higher, independent of patient age or weight at surgery; the reason remains unclear. However, in the present study, the presence of Down syndrome was not associated with an increased rate of new arrhythmias after surgery. Regrettably, we performed statistical analysis addressing arrhythmias as a whole, and not specifically looking at AV block. This result is therefore surprising and probably simply due to a relevant methodological limitation of our study, which was based on the limited data available in the national database we analyzed.

In the present study, patients with Down syndrome had a longer hospital stay after surgery, consistent with previous studies [16,19], suggesting a higher infection rate in the postoperative period as well as a plethora of other causes; for example, pulmonary hypertension, feeding problems, muscular hypotonia, obstructive sleep apnea syndrome, and thyroid dysfunction in the perioperative stressful period, younger age at operation, less experienced parents, and more comorbidities [19,23].

This registration-system-based study has several limitations. Because data are recorded for the Korean fee-for-service payment and reimbursement system, limitations may exist in the accuracy and precision of ICD-10 coding used by participating institutions as well as limitations in the clinical details of patient encounters that are provided based on ICD diagnostic and procedural coding. Furthermore, the database only contains selected data, so that a number of relevant characteristics could not be appropriately analyzed (for example, we did not know and could not analyze the causes of hospitalization or of emergency department visits). The study cohort consisted of Down syndrome patients with congenital heart disease who underwent corrective surgery and did not include Down syndrome patients who did not undergo corrective surgery. There could be various reasons for not undergoing corrective surgery, including no need for surgery, refusing surgery, and only palliative surgery. Therefore, patients with a serious condition may have been missed. The clinical characteristics of the cohort in this database cannot be described as preoperative or postoperative, limiting the evaluation of perioperative variables for outcomes. Although a single trisomy 21 patient can have multiple congenital heart defects, only one single diagnosis is assigned to every patient in the national database, according to priority rules acknowledging the degree of complexity and influence on later life. Such unacknowledged diagnoses could be a further cause of limitation. The strength of our research lies in compensating for these shortcomings. First, sensitivity analyses were conducted, and all supported the main findings. Second, since retrospective national cohort studies are limited by various sources of biases and confounders, IPTW of the propensity score was performed. Through propensity score adjustment, researchers can account for comparability between groups. Finally, a relevant strength of this study is its nationwide coverage by balancing the distribution of bias and confounding factors between groups.

## 5. Conclusions

Following total corrective surgery in Down syndrome children with congenital heart disease and in a control group during the same period, higher late mortality and longer postoperative hospitalization were observed in the Down syndrome group. Pulmonary hypertension occurred more often in Down syndrome patients after total correction. However, similar rates were observed for heart failure and arrhythmias in the Down syndrome and control groups. In conclusion, this study shows that, despite many medical advances, the mortality rate of patients with congenital heart disease is increased in the presence of Down syndrome.

## Figures and Tables

**Figure 1 children-09-01329-f001:**
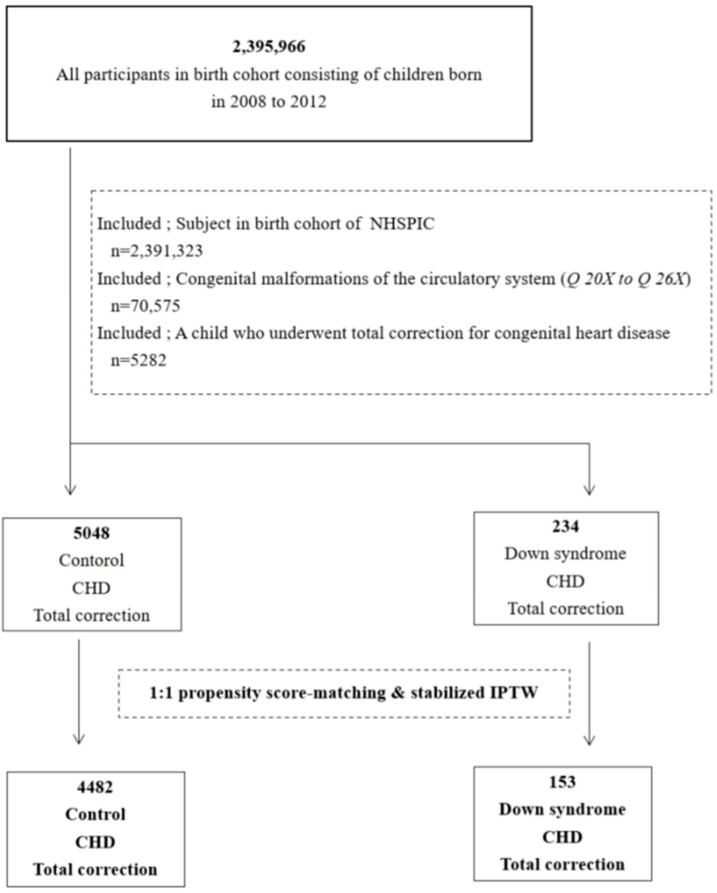
Flowchart of participant selection from the birth cohort. Among the 70,575 children with congenital heart disease, 5282 who underwent total correction were included in the subject group; 234 had Down syndrome and the remaining 5048 were children without Down syndrome. Propensity score matching (maximum 1:1) and stabilized IPTW were performed to assign 153 participants to the Down syndrome group and 4482 participants to the non-Down syndrome group. NHSPIC = National Health Screening Program for Infants and Children, CHD = congenital heart disease, IPTW = inverse probability treatment weighting.

**Figure 2 children-09-01329-f002:**
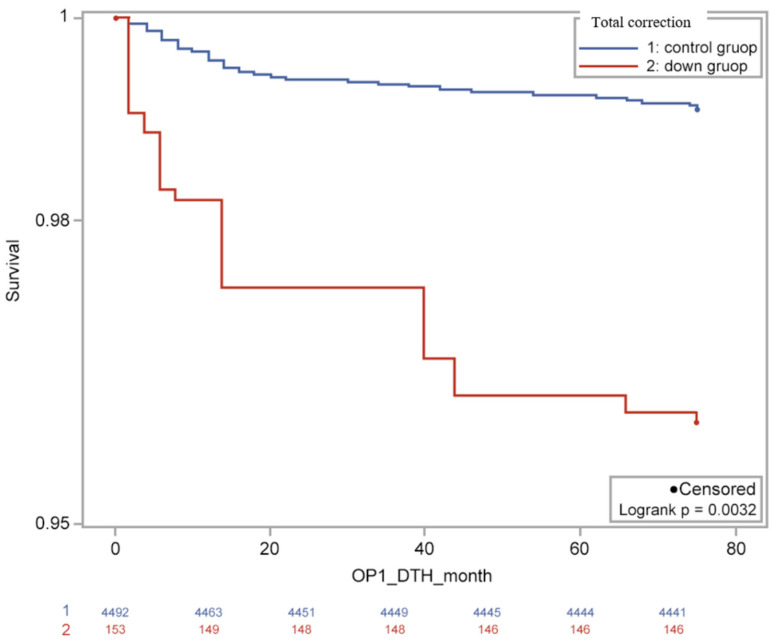
Kaplan–Meier survival curve in the Down syndrome and control subjects who underwent total correction. The cumulative survival rate in the control group was significantly higher than in the Down syndrome group (HR = 4.3, 95% CI: 1.9–9.9). HR, hazard ratio; CI, confidence interval.

**Table 1 children-09-01329-t001:** Baseline demographic characteristics of the participants with and without Down syndrome who underwent total correction for congenital heart disease.

Demographic Characteristics	Observed Data (N = 5282)			Propensity-Matched Data (N = 4635)		
	Total Correction for CHD, N (%) ^a^			Total Correction for CHD, N (%) ^a^		
	Control	Down	SMD ^b^	Control	Down	SMD ^b^
Total, n	5048	234		4482	153	
Male	2641 (52.3)	126 (53.8)	−0.02	2343 (52.3)	74 (48.4)	0.07
Female	2407 (47.7)	108 (46.2)		2139 (47.7)	79 (51.6)	
Prematurity, n ^c^	433 (8.5)	24 (10.30)	0.10	437 (9.8)	22 (14.4)	0.14
Birth weight ^d^, kg (IQR) ^e^	3.1 (2.7–3.4)	2.9 (2.5–3.2)	−0.29	3.0 (2.7–3.4)	2.9 (2.6–3.2)	−0.17
Birthplace, n ^f^						
Seoul	1059 (21.0)	50 (21.4)	0.03	926 (20.7)	30 (19.6)	−0.02
Metropolitan area	1178 (23.3)	54 (23.1)	−0.04	1049 (23.4)	34 (22.2)	−0.02
Urban area	2338 (46.3)	104 (44.4)	−0.04	2084 (46.5)	76 (49.7)	0.06
Rural	421 (8.3)	25 (10.7)	0.10	377 (8.4)	11 (7.2)	−0.02
Economic status, n ^g^						
1st _(low)_	419 (8.3)	22 (9.4)	0.07	364 (8.1)	10 (6.5)	−0.06
2nd	722 (14.3)	33 (14.1)	0.02	632 (14.1)	21 (13.7)	−0.01
3rd	1301 (25.8)	63 (26.9)	0.02	1182 (26.4)	45 (29.4)	0.07
4th	1524 (30.2)	54 (23.1)	−0.16	1363 (30.4)	45 (29.4)	−0.02
5th	873 (17.3)	50 (21.4)	0.04	757 (16.9)	26 (17.0)	0.01
Year of birth						
2008	925 (18.3)	44 (18.8)	−0.00	763 (17.0)	23 (15.0)	−0.05
2009	917 (18.2)	46 (19.7)	−0.01	808 (18.0)	29 (19.0)	0.02
2010	1006 (19.9)	40 (17.1)	−0.04	900 (20.1)	33 (21.6)	0.04
2011	1070 (21.2)	53 (22.6)	0.03	984 (22.0)	33 (21.6)	0.00
2012	1130 (22.4)	51 (21.8)	0.03	1027 (22.9)	34 (22.2)	−0.01
Surgery						
Total correction, n	4521 (88.9)	217 (92.7)	0.14	4042 (90.2)	141 (92.2)	0.06
Palliation and total correction, n	527 (10.4)	17 (7.3)	−0.14	440 (9.8)	12 (7.8)	−0.06
Age at total correction, months (IQR) ^e^	4 (1–11)	4 (2–7)	−0.24	4 (1–11)	5 (2–8)	−0.18

^a^ Values are reported as a number (%) unless otherwise indicated. n, number of patients; CHD, congenital heart disease; SD, standard deviation. ^b^ SMD, standardized mean difference; a value >10% was considered a significant difference. ^c^ The prematurity status data obtained by parental-reporting general questionnaire of the National Health Screening Program of Infants and Children. An infant was considered premature if born before gestational age of 37 weeks. ^d^ The birth weight data were obtained by parental-reporting questionnaire of the National Health Screening Program of Infants and Children. ^e^ Categorical variables were expressed as frequency distributions, and continuous variables as median and IQR. ^f^ Metropolitan areas were defined as Busan, Incheon, Gwangju, Daejeon, and Ulsan; urban areas as other cities, and rural areas as non-city areas (“gun”) (birthplace missing in observed data = 2.4%). ^g^ Economic status was divided into quintiles based on insurance contribution (economic status in observed data = 4.6%).

**Table 2 children-09-01329-t002:** The preoperative distribution of congenital heart disease in subjects with and without Down syndrome.

CHD	Priority	Total (N = 5282)	Control (N = 5048)		Down Syndrome (N = 234)		
		n	n	%	n	%	*p*-Value
AVSD *	1	674	585	11.5	89	38.0	<0.0001
TOF	2	863	844	16.7	19	8.1	0.000
TGA	3	398	396	7.8	2	0.8	<0.0001
SV	4	170	170	3.3	0	0.0	0.004
CoA	5	353	346	6.8	7	2.9	0.020
DORV	6	119	113	2.2	6	2.5	0.742
VSD	7	1731	1640	32.4	91	38.8	0.041
PA	8	44	43	0.8	1	0.4	0.484
AS	9	27	27	0.5	0	0.0	0.262
PS	10	113	112	2.2	1	0.4	0.064
PDA	11	142	137	2.7	5	2.1	0.593
ASD	12	581	568	11.2	13	5.5	0.006
TAPVR and PAPVR	13	46	46	0.9	0	0.0	0.142
OTHERS	14	21	21	0.4	0	0.0	0.322

CHD, congenital heart disease; AVSD, atrioventricular septal defect; TOF, tetralogy of Fallot; TGA, transposition of great arteries; SV, single ventricle (double inlet ventricle, hypoplastic right heart syndrome, hypoplastic left heart syndrome); CoA, coarctation of aorta; DORV, double outlet right ventricle; VSD, ventricular septal defect; PA, pulmonary atresia; AS, aortic valve stenosis; PS, pulmonary stenosis; PDA, patent ductus arteriosus; ASD, atrial septal defect; TAPVR and PAPVR, total anomalous pulmonary venous return and partial anomalous pulmonary venous return. * AVSD patients not amenable to corrective surgery and being treated with a single ventricle palliation were not included in this series.

## Data Availability

The datasets used and/or analyzed during the current study are available from the corresponding author upon reasonable request.

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
