# Peer review of "Long-Term Observational Outcomes after Total Correction of Congenital Heart Disease in Korean Patients with Down Syndrome: A National Cohort Study"

_children, 2022, doi:10.3390/children9091329_

Round 1
Reviewer 1 Report
The manuscript shows the real-world data of the patients with CHD and Down's syndrome in Korea. Although the novelty is not great, the analysis is highly suggestive based on large-scale nationwide data. However, there are some points to be revised.
1. (Abstract) The sentence "Down syndrome was the most common chromosomal... " should be placed in the result section.
2. (Material and Methods) Your data of BMI and height seems to be analyzed in the manner of percentile. I think the word "z-scores" is not needed in this case. Please consult the statistician on the appropriate word choice.
3. (Result) "Birth defects occurred in 1243 of the total 2395966 with a prevalence of 5.18 per 10000". What does it mean?
4. (Result) There is inappropriate space 'p resented' in line 192.
5. (Figure2) The number of Down group (D should be capitalized) is 185. I think it should be the comparable number of propensity-matched data 153.
6. (Discussion) In the description"the weight percentile was similar in both..." , weight should be replaced with BMI. In case the BMI is similar and height is different, the weight should be different. Please clarify.
7. (Discussion) What does the difference in the ER visit mean?
Author Response
22nd . July, 2022
Prof. Dr. Sari A. Acra
Editor-in-Chief
Children
Dear Dr. Sari A. Acra
I, along with my coauthors, would like to re-submit the attached manuscript entitled “Long-term observational outcomes after total correction of congenital heart disease in Korean patients with Down syndrome: A national cohort study” as an original article. The manuscript ID is children-1812650
The manuscript has been carefully rechecked and appropriate changes have been made in accordance with the reviewers’ suggestions. The responses to their comments have been prepared and attached herewith.
We thank you and the reviewers for your thoughtful suggestions and insights, which have enriched the manuscript and produced a more balanced and better account of the research. We hope that the revised manuscript is now suitable for publication in your journal.
I look forward to your reply.
Sincerely,
Jinyoung Song
Department of Pediatrics, Samsung Medical Center
Sungkyunkwan University, School of Medicine
50 Irwon-dong, Gangnam-gu, Seoul, Korea 135-710
82-2-3410-3539

Reviewer 2 Report
This study based on the national investigation of birth cohort in Korea. demonstrated that Down syndrome patients with congenital heart disease undergoing corrective surgery showed residual pulmonary hypertension after corrective surgery, longer duration of hospitalization, and frequent hospitalizations after corrective surgery, and higher late mortality. There are several comments as below.
Abstract:
Abstract was appropriately described according to the context. Conclusions are better to be rephrased to “Down syndrome patients with congenital heart disease undergoing total correction showed residual pulmonary hypertension after surgery, longer length of hospitalization, frequent hospitalization after surgery, and higher late mortality.”
Introduction:
Introduction was clearly described to state the aim of this study. Line66; “and total correction after staged palliation” can be delated.
Material and Methods:
The authors appropriately described study population and methods for analysis. In addition, the ethical statements were clearly described.
Results:
In “3.2 Preoperative distribution of congenital heart disease”, the study cohort consisted of DS patients with congenital heart disease who underwent corrective surgery, which did not involve DS patients who did not require surgery. Therefore, the authors have better to state this issue in the Discussion or Limitation section.
Survival rates analyzed by the Kaplan-Meier method usually involve a time domain. Although the authors described that the Kaplan-Meier method was used to compare the survival rates between the groups using log-rank tests in the Method section, the survival rates shown as 8.12% (DS groups) vs. 3.65% (control group) were likely to be calculated as the number of deceased patients divided by the total number of a cohort. Please explain this issue.
Do the authors have available data regarding home oxygen therapy or other medications after corrective surgery?
Discussion:
The authors appropriately described Discussion point by point.
Tables and figures were sufficient.
Author Response

(The authors gave the same response as above.)
